# Dietary Fiber in Bilberry Ameliorates Pre-Obesity Events in Rats by Regulating Lipid Depot, Cecal Short-Chain Fatty Acid Formation and Microbiota Composition

**DOI:** 10.3390/nu11061350

**Published:** 2019-06-15

**Authors:** Hao-Yu Liu, Tomas B. Walden, Demin Cai, David Ahl, Stefan Bertilsson, Mia Phillipson, Margareta Nyman, Lena Holm

**Affiliations:** 1Department of Medical Cell Biology, Uppsala University, 75123 Uppsala, Sweden; tomas.walden@mcb.uu.se (T.B.W.); david.ahl@mcb.uu.se (D.A.); mia.phillipson@mcb.uu.se (M.P.); lena.holm@mcb.uu.se (L.H.); 2Department of Biochemistry and Molecular Medicine, University of California at Davis, Sacramento, CA 95817, USA; 3Department of Ecology and Genetics, Limnology and Science for Life Laboratory, Uppsala University, 75236 Uppsala, Sweden; Stebe@ebc.uu.se; 4Department of Food Technology, Engineering and Nutrition, Lund University, 22100 Lund, Sweden; margareta.nyman@food.lth.se

**Keywords:** adipose tissue, bilberry, butyrate-producing bacteria, gut microbiota, hepatic steatosis, lipid metabolism, obesity-resistant, fermentation, prebiotic dietary fiber, short-chain fatty acids

## Abstract

Obesity is linked to non-alcoholic fatty liver disease and risk factors associated to metabolic syndrome. Bilberry (*Vaccinium myrtillus*) that contains easily fermentable fiber may strengthen the intestinal barrier function, attenuate inflammation and modulate gut microbiota composition, thereby prevent obesity development. In the current study, liver lipid metabolism, fat depot, cecal and serum short-chain fatty acids (SCFAs) and gut microbiome were evaluated in rats fed bilberries in a high-fat (HFD + BB) or low-fat (LFD + BB) setting for 8 weeks and compared with diets containing equal amount of fiber resistant to fermentation (cellulose, HFD and LFD). HFD fed rats did not obtain an obese phenotype but underwent pre-obesity events including increased liver index, lipid accumulation and increased serum cholesterol levels. This was linked to shifts of cecal bacterial community and reduction of major SCFAs. Bilberry inclusion improved liver metabolism and serum lipid levels. Bilberry inclusion under either LFD or HFD, maintained microbiota homeostasis, stimulated interscapular-brown adipose tissue depot associated with increased mRNA expression of uncoupling protein-1; enhanced SCFAs in the cecum and circulation; and promoted butyric acid and butyrate-producing bacteria. These findings suggest that bilberry may serve as a preventative dietary measure to optimize microbiome and associated lipid metabolism during or prior to HFD.

## 1. Introduction

Obesity and diseases related to the metabolic syndrome, such as cardiovascular diseases, type 2 diabetes and non-alcoholic fatty liver disease (NAFLD) are ever-growing and have become epidemic worldwide [1]. Several studies have demonstrated that diets high in fat may induce gut microbiota dysbiosis, which is believed to be an important contributor to the onset of obesity and related diseases [2,3,4], possibly through changes of bile acid synthesis in the liver that secret into the gut [5]. Consequently, the host energy balance is disrupted, as high-fat associated gut microbiome has an increased capacity to harvest energy from the diet, leading to obesity [2,3,6]. Furthermore, a high-fat diet induces changes of the intestinal bacterial community with outburst of some potential pathogens, resulting in impaired gut barrier function, systemic low-grade inflammation and endotoxin in blood [7].

The distal gastrointestinal (GI) tract of mammals contains a complex and dynamic microbial community that are shaped by environmental factors, such as the diet [3]. The microbiota may affect a range of functions in the host, including alterations of the immune system and the metabolism [3,7,8,9]. Feeding strategies where the dietary fiber are enriched are thus developed to improve gut microbiota homeostasis and to manage disorders related to the metabolic syndrome [10,11]. Dietary fibers escape the small intestinal digestion and reach the large intestine where they provide adherent site for the microbiota and constitute energy. Some types of dietary fibers are highly degraded by the microbiota, while others provide physical benefits such as increased fecal bulking and laxation [12,13,14]. If increasing the growth and activity of beneficial bacterial species in the host, dietary fibers are referred to as prebiotics [11,15]. The most well-known food components having prebiotic properties are the fructans, which are known to selectively stimulate the growth of *Bifidobacteria* and *Lactobacilli* [16,17]. This selective bacterial promotion may result in an enhanced gut barrier function, and protect against systemic inflammation, liver disruption and insulin resistance [15].

Following microbial degradation, short-chain fatty acids (SCFAs) are formed and determined by the source and structure of fibers to affect gut health [9,10,15,18]. The SCFAs pattern formed is also affected by the dietary fat content via the bile-acids [10,19,20]. The SCFAs comprise mainly acetic, propionic and butyric acids. These metabolites represent a wide range of beneficial actions, since they are absorbed and thus can be detected in the circulation [9,18,21]. Acetate is the most abundant SCFA found in the large intestine and is acting as a substrate for hepatic cholesterol synthesis, while propionate is reported to have both positive and negative effects. It contributes to gluconeogenesis locally in the intestine [9], and is also shown to increase adipogenesis and inhibit lipolysis in mice [22]. On the other hand, propionate may decrease liver fatty acid content and improve insulin sensitivity, providing benefits in the context of prevention of obesity and type 2 diabetes, as shown in mice [12]. In this respect, the ratio between acetic- and propionic acid can be an important indicator [5]. Finally, butyrate is the most important energy substrate for the colon enterocytes and is therefore suggested to play a role in maintaining intestinal barrier functions [18], as well as in shaping the host metabolism. Indeed, it has been demonstrated that cereal products rich in fiber rapidly affect microbiota composition (15 h postprandially) and increase butyric acid formation in healthy subjects, while lower their glucose level and induce higher appetite responses [23]. It is noteworthy that butyrate has been reported to exert antisteatotic hepatic effects via direct AMP kinase stimulation in diabetic mice [24]. Thus, the control of microbiota fermentation and lipid metabolism in the intestine or beyond seems to be of vital importance for host health.

Interestingly, several studies have described a naturally occurring ‘obesity-resistant’ phenotype within a single litter of inbreed rats or mice [25,26]. The ‘obesity-resistant’ individuals are unable to gain weight to the same extent as their ‘obesity-prone’ siblings and they do not develop metabolic symptoms such as low-grade inflammation and insulin resistance. Although it is well established that diet induced obesity is associated with microbiota alteration, few data are available regarding animals and humans that are resistant to obesity under a high-fat diet regimen and their microbiota signatures [26,27,28].

Bilberries (*Vaccinium myrtillus*) contain easily fermentable dietary fibers and could thus be recognized as a prebiotic food item, and have been shown to improve host health through anti-oxidant and anti-inflammatory actions [11,29,30]. The aim of the present study was to evaluate whether metabolic events could be related to occurrences in the large intestine of rats after administration of bilberries containing high amounts of easily fermentable fibers in a low- or high-fat setting. Bilberry diets were compared with diets containing cellulose with an equal content of dietary fiber that was insoluble and resistant to microbial degradation in the cecum. Parameters investigated were liver lipid metabolism, fat depots, cecal and serum SCFAs and cecal microbiome in rats.

## 2. Materials and Methods

### 2.1. Animals and Diets

Thirty-two male Sprague–Dawley rats weighing 70–80 g (4 weeks old) were obtained from Taconic Farms (Ejby, Lille Skensved, Denmark) and maintained in a temperature- and humidity-controlled room (23 °C ± 1 °C and relative humidity of 60 ± 5%) with a 12 h light–dark cycle. The rats were randomly divided into four groups (n = 8) and fed either a low-fat (LFD) or high-fat diet (HFD) containing cellulose (microcrystalline, MCC, FMC BioPolymer Cork, Cork, Ireland) (7% dietary fiber, on dry weight basis, dwb), or corresponding diets containing 7% (dwb) dietary fiber from bilberries (BB) (*Vaccinium myrtillus*), LFD + BB or HFD + BB diet, respectively (Figure 1). The diets contained the same amount of protein (12%), as casein plus DL-methionine, sucrose (10%, dwb), rapeseed oil (5%, dwb), minerals (4.8%, dwb) and vitamins (1%, dwb). Wheat starch was used to adjust dry matter content and is completely digested. The bilberries (Kiviks Musteri AB, Kivik, Sweden) were frozen when obtained, freeze-dried and milled to a particle size less than 0.5 µm (Cyclotec, Höganäs, Sweden) before included in the diets. To give 7% dietary fiber (dwb), 29% bilberries had to be included in the diets and the rats were fed the diets for 8 weeks. Diets and water were supplied ad libitum. On the last day of the experiment, animals were anesthetized with 120 mg/kg body weight of 5-ethyl-5-(1-methylpropyl)-2-thiobutabarbital sodium (Inactin^®^, Sigma, St. Louis, MO, USA) given intraperitoneally. Blood (serum) was collected from a. carotis and v. porta and was frozen at −80 °C until analysis. Spleen, cecum, liver and adipose tissues of three compartments, i.e., visceral epididymal white adipose tissue (eWAT), subcutaneous inguinal white adipose tissue (iWAT) and interscapular brown adipose tissue (iBAT) were collected and weighed. The adipose tissues were directly snap frozen in liquid nitrogen and the liver was also embedded in Neg50 O.C.T. (optimal cutting temperature)-medium (Cellab, Stockholm, Sweden) for immunohistochemistry analysis. The cecum, cecal contents were sampled, snap-frozen in liquid nitrogen and stored at −80 °C until analysis of SCFAs and microbiota composition.

Experiments were approved by the Swedish Laboratory Animal Ethical Committee in Uppsala in accordance with animal experiment number C344/12 and all procedures were conducted according to the Swedish National Board for Laboratory Animals guidelines. The rats appeared healthy and active throughout the experiment and the diets were well tolerated. The animals were weighed weekly.

#### Dietary Fiber Analyses

Fresh frozen bilberries from Kiviks Musteri (Kivik, Sweden) were freeze-dried and milled before the dietary fiber content was determined by an enzymatic-gravimetric AOAC methodology developed by Prosky and collaborators [31]. The main dietary fiber components of bilberries used in this study were xyloglucans in both the soluble and insoluble fraction, and thirteen different derivatives of anthocyanins. The content of anthocyanins in bilberries was quantified to be 233 μg/g (dwb) in the soluble fraction and 1559 μg/g (dwb) in the insoluble fraction, i.e., 4–5 times higher than in corresponding segments from raspberry and black currant analyzed by quadrupole LC-MS with electrospray ionization (Agilent 1260 Infinity system) [32]. The number of anthocyanins was also higher in bilberries than in the two other berries [10,32]. In addition, xyloglucans in the soluble fraction were more branched and contained comparatively more galactose and to some extent also arabinose, highly degradable to the microbiota [32].

### 2.2. Liver Histology

Liver O.C.T.-samples were sectioned (10 µm thick) at two different levels approximately 60 µm apart using a cryostat, Leica CM1860 UV (Leica Microsystems, Solms, Germany), n = 5, (duplicate slides per individual). Sections were stained with Oil Red O (ORO) and hematoxylin (Abcam product #ab150678). Six images taken at different areas of each slide at 10× magnification were obtained using a Leica DFC420C camera for each sample. Images were analyzed using the software Image Processing and Analysis in Java (ImageJ, National Institute of Health, Madison, WI, USA) where percent ORO-positive area was calculated per image area to give the relative staining level for each sample.

### 2.3. RNA Isolation and Real-Time qPCR

Total RNA was extracted from liver and adipose tissues with Trizol (Life Technologies, Carlsbad, CA, USA) according to the manufacturer’s protocol, and RNA concentrations were measured on a Nanodrop™ 2000c spectrophotometer (Thermo-Scientific, Wilmington, DE, USA). To synthesize cDNA, 2µg RNA from each sample were reverse-transcribed with a High Capacity cDNA kit (Life Technologies, Carlsbad, CA, USA) in a total volume of 20 µL and diluted to 60 µL. To measure mRNA expression, 1 µL of each cDNA sample was loaded in duplicate. The SsoFast EvaGreen Supermix™ (Bio-Rad) master-mix together with pre-validated, designed primers (Universal Probe Library, Roche Applied Science, Penzberg, Germany) was used (Appendix A). Expressions of fatty acid synthase (*Fasn*), acetyl CoA carboxylase (*Acc*), fatty acid binding protein 5 (*Fabp5*), stearoyl-CoA desaturase-1 (*Scd-1*), microsomal triglyceride transport protein (*Mttp*), peroxisome proliferator-activated receptor α and γ (*Pparα* and *Pparγ*), carnitine palmitoyl transferase 1 α (*Cpt1α*), uncoupling protein-1 *(Ucp-1*) and transcription factor II B (*TfIIb*) were studied. The real-time qPCR measurements were performed on a MyIQ2 Real-Time PCR system in a 96 well format (Bio-Rad). All mRNA levels were normalized to the endogenous control *TfIIb* expression and presented as 2^−ΔΔCt^ using average of the LFD group as 100.

### 2.4. Total Cholesterol, Triglycerides, and Free Fatty Acids

Serum concentrations of total cholesterol (TCHO), triglycerides and free fatty acids were determined using the standard Schmid-Bondzynski-Ratzlaff method [33] with the Architect c4000 analyzer (Abbott Diagnostics Abbott Park, IL, USA).

### 2.5. Carboxylic Acids

SCFAs were extracted from cecal content (~2 g) and analyzed by using gas-liquid chromatography (GLC) as described previously [34,35]. Briefly, the cecal content was homogenized and protonated with hydrochloric acid, by using 2-ethylbutyric acid as internal standard. Samples were centrifuged before injection onto a fused-silica capillary column (DB-FFAP 125-3237; J&W Scientific, Agilent Technologies Inc., Folsom, CA, USA). SCFAs in serum were pre-enriched and extracted by hollow-fiber before being injected and analyzed with GLC using the same column as for SCFAs extracted from cecum. GC ChemStation software (Agilent Technologies Inc.,Wilmington, DE, USA) was used for evaluation of the SCFA analysis. For analysis of lactic and succinic acids ion-chromatography was employed according to Jakobsdottir et al. [10]. The same extraction as for SCFAs was used.

### 2.6. DNA Extraction and 16S rRNA Gene Illumina Sequencing

DNA was extracted from 180–220 mg of cecal content, using QIAmp DNA Stool Mini kit (Qiagen, Hilden, Germany). The procedure was carried out according to the manufacturer’s instruction after homogenizing the samples with bead beating twice at speed six using MP FastPrep-24 (MP Biomedicals).

For Illumina MiSeq Dual Index Amplicon sequencing, the bacterial V3-V4 16S rRNA gene regions were first PCR amplified from each sample using a composite forward primer and a reverse primer, each containing a unique eight base index primer, designed to tag PCR product from the respective samples. The forward primer was Illumina adapter-N4-341F:3′-ACACTCTTTCCCTACACGACGCTCTTCCGATCTNNNNCCTACGGGNGGCWGCAG-5′. The reverse primer was Illumina adapter-805R:3′-AGACGTGTGCTCTTCCGATCTGACTACHVGGGTATCTAATCC-5′. PCR reactions consisted of master mix (0.5 µL of each primer, 4 µL of Q5 reaction buffer, 0.2 µL (0.02U µL-1) Q5 High-Fidelity DNA Polymerase (New England Biolabs, Ipswich, MA, USA), 2 µL of dNTPs [Deoxynucleotide], 11.8 µL of Milli-Q water) and 1 µL of DNA template. Reaction conditions were 1 min at 98 °C, followed by 20 cycles of 10 s at 98 °C, 30 s at 58 °C and 30 s at 72 °C, and a final extension step at 72 °C for 2 min on a Bio-Rad thermocycler. Duplicates were run in 20 μL reactions for each sample, combined, purified with Agencourt Ampure magnetic purification beads (Beckman Coulter, Bromma, Sweden). Meanwhile, products were visualized by gel electrophoresis. Next, a second PCR was conducted using the first PCR product for attaching standard Illumina handles and index primers. Finally, the PCR products were quantified with Picogreen dsDNA assay according to the manufacturer’s instruction (Thermo Fisher Scientific, Waltham, MA, USA) in order to pool equimolar amounts of amplicon for sequencing. The master DNA pool was generated and pair-end sequencing using Miseq Reagent kit v3 with read lengths of 2 × 250 bp was performed on a MiSeq sequencer (Illumina) at the SciLifeLab SNP&SEQ sequencing facility (Uppsala, Sweden).

The Illumina sequencing data output was processed according to the cut-offs and pipeline customized by Sinclair et al. [36]. Sequences were assigned to operational taxonomic units (OTUs) by using a closed reference-based OTU picking method in QIIME v1.8. For every OTU, the sequence was checked as a query against the SILVAMOD database using the CREST software version 2.0. The sequences were aligned to the GenBank database using standard nucleotide BLAST at NCBI (http://www.ncbi.nlm.nih.gov). Bioinformatics analysis was performed using the cumulative-sum scaling method where raw data was divided with the cumulative sum of counts up to a percentile assessed by a data-driven approach [37]. The bacterial 16S rRNA amplicon sequence data are available in European Nucleotide Archive database under accession number PRJEB12147 (ERP013589).

### 2.7. Statistical Analysis

Differences between the four experimental groups were assessed by ANOVA corrected for multiple comparisons with Tukey’s post-hoc test using GraphPad Prism version 6.07. Interaction between high fat and bilberry factor was tested without any significance. To obtain the cecal pool of SCFAs (µmol), the concentration of SCFAs (µmol/g) in the cecal content was multiplied by sample weight. For total SCFAs, ANOVA with Contrast analysis was used. Values are expressed as mean ± SEM (stand error of the mean). Statistics for the Illumina sequencing of bacterial 16S rRNA amplicons were performed using version 2.0-7 of the VEGAN package and the R statistical framework version 2.11, including multidimensional scaling analysis, alpha-diversity and ANOVA with false discovery rate control. For all analyses, *p* < 0.05 is considered as significant.

## 3. Results

### 3.1. Body and Tissue Weights

No differences in final body weight (BW) was observed between the HFD group and the LFD and LFD + BB groups (Appendix A, *p* > 0.05) after 8 weeks feeding. However, the final BW was less in the group fed HFD + BB than in the other groups (Appendix A, *p* < 0.05–*p* < 0.01). Furthermore, diets with bilberry increased the cecum weights (including both the content and tissue) in rats, compared with those fed LFD and HFD, respectively (Appendix A, *p* < 0.001). There were no differences in the spleen weights (g/kg BW, data not shown).

### 3.2. Lipid Metabolic Profiles in the Liver

Rats fed HFD exhibited a significantly higher liver index (calculated as liver weight g per kg BW) compared to rats with other treatments (Figure 2A, *p* < 0.05). This was consistent with the observations that heavier livers from HFD were also visually paler (Appendix A). More importantly, liver histology with ORO staining indicating steatosis, revealed a dramatic increase of lipid accumulation in HFD fed rats (43.8 ± 2.24%) in contrast to rats fed LFD (2.2 ± 0.22%) and LFD + BB (2.8 ± 0.27%) groups (Figure 2B,C). Livers in rats fed HFD + BB had significantly lower lipid accumulation than rats fed only HFD, but the ORO positive staining was higher than in rats fed LFD and LFD + BB (Figure 2B,C, *p* < 0.001). In contrast to the lipid staining, both high-fat diets down regulated expression of genes related to *de novo* lipogenesis, including *Fasn*, *Fabp5*, *Acc* and *Scd-1* when compared to LFD control and LFD + BB groups (*p* < 0.05) (Figure 2D). A decreased expression was also found on fatty acid transport and lipoprotein assembly genes *Fabp5* and *Mttp* at mRNA level in response to HFD (Figure 2D, *p* < 0.05), while there was no difference in the expression of genes involved in liver β-oxidation, *Pparγ*, *Pparα* and *Cpt1α* between diets (*p* > 0.05). These data led us to investigate lipid and energy changes elsewhere in the rat body in response to HFD with or without bilberries.

#### 3.2.1. Adipose Tissues

HFD with or without bilberries had similar adipose tissue (eWAT or iWAT) distribution (Figure 3A, *p* > 0.05). In contrast, LFD+BB had significantly lower eWAT mass compared to the LFD group (*p* < 0.05), while iBAT mass was significantly higher in rats fed HFD + BB than all the other diets (*p* < 0.05). In addition, gene expression of *Ucp-1*, an indicator of sympathetic stimulation of brown adipose tissue associating with adaptive thermogenesis, was consistent with adipose tissue mass changes, whereas the expression in eWAT or iWAT was not affected by dietary treatments (Figure 3B, *p* > 0.05). The mRNA level in iBAT was significantly higher in rats fed HFD+BB than all the other diets (*p* < 0.01), suggesting that bilberry had a strong effect on adipose tissue, independently of the amount of fat inclusion.

#### 3.2.2. Lipid Levels

The serum total cholesterol (TCHO) levels were significantly higher in rats fed HFD compared to rats fed LFD (*p* < 0.05) and LFD + BB (Figure 4, *p* < 0.001). HFD + BB counteracted this increase (*p* < 0.01). However, serum triglycerides and free fatty acids were found at similar levels in all diet groups (Figure 4, *p* > 0.05).

### 3.3. SCFAs in Cecum and Serum

The total cecum and serum SCFAs concentrations increased significantly by bilberries (Appendix A, Figure 5A,E, *p* < 0.05), while the cecal formation of total SCFAs was markedly reduced by high-fat feeding (Appendix A). The main cecal SCFAs, acetic, propionic and butyric acids, showed similar trends (Appendix A), but the cecal formation of butyric acid with bilberries was independent of the amount of fat in the diet and formed similar amounts in rats fed with low and high-fat diets (Figure 5D and Appendix A). The total SCFAs in serum was not reduced compared to the LFD groups and the amounts were similar with high- and low-fat feeding, except for propionic acid where the amount decreased in the high-fat setting (Figure 5E). The amount of succinic acid increased in the rats with both bilberry diets (LFD + BB and HFD + BB) compared to the LFD group, while lactic acid formation was similar among groups (Appendix A, *p* > 0.05). Consequently, changes of SCFAs in cecum and serum were mainly attributed to the dietary fiber components.

### 3.4. Microbiota Dynamics and Community Compostion

The gut microbiota serves as an essential nexus between dietary components fermentation and the SCFAs formation. By using next generation sequencing of 16S rRNA gene amplicons based on OTUs, we revealed distinct taxonomical profiles of LFD, LFD + BB, HFD and HFD + BB-associated microbiome with multidimensional scaling analysis. The bacterial community was shifted in response to dietary factors i.e., fat content and bilberry inclusion (Figure 6A). Furthermore, the α-diversity (calculated as Shannon index) of cecal microbiota was significantly lower in the HFD group as compared to the low-fat settings (LFD and LFD + BB, Figure 6B, *p* < 0.05), indicating a dysbiotic state of the bacterial community. In contrast, bilberry inclusion in the high-fat setting (HFD + BB) was able to normalize the α-diversity (*p* < 0.001). In accordance with previous studies [38], the rat microbiota was dominated by the phyla *Firmicutes* and *Bacteroidetes*, where the relative abundance of *Firmicutes* was significantly higher in HFD group compared to LFD (Figure 6C, *p* < 0.05). Important changes at the phylum level, by the HFD, also included a decrease in *Verrucomicrobia* and an increase in *Proteobacteria* (*p* < 0.05). Interestingly, bilberry treatment (HFD + BB) exhibited the highest relative abundance of *Actinobacteria* (*p* < 0.05) and inhibited the *Firmicutes* increase (*p* < 0.05).

The relative abundance of top-ranking genera in cecal microbiota was significantly affected by LFD, LFD + BB, HFD or HFD + BB diet (Appendix A) when adjusted by false discovery rate (FDR) control. Comparing bacterial community of different diets with LFD fed rats, HFD driven changes of microbiota were represented by *Clostridium* (*Firmicutes*), *Peptostreptococcaceae Incertae Sedis* (*Firmicutes*), *Lachnospiraceae Incertae Sedis* (*Firmicutes*) and *Escherichia-Shigella* (*Proteobacteria)* (*p* < 0.05), with the latter three known as potential pathogens. Furthermore, there was a loss of *Akkermansia* sp., represented by the decrease of phylum *Verrucomicrobia* (*p* < 0.05). Interestingly, bilberry treatment under either LFD or HFD, favored the representation of *Bacteroides* belonging to the phylum *Bacteroidetes* in the cecal bacterial community (*p* < 0.05) and there was a fivefold increase of *Actinobacteria* phylum (Figure 6C, *p* < 0.05) in HFD + BB fed rats with the most significant changes from the genus *Bifidobacteria* (Appendix A, Figure 6D, *p* < 0.0001). HFD feeding also increased the relative abundance of this bacteria in the rats compared to LFD and LFD + BB groups (Figure 6D, *p* < 0.001). In this study, it is noteworthy that the relative abundance of a group of butyrate-producing bacteria (i.e., *Roseburia*, *Anaerotruncus*, *Coprococcus* and *Anaerostipes*) was increased when bilberries were added in both LFD and HFD (Figure 6E, *p* < 0.01), which may explain diet induced increases of butyric acids in the cecal content and serum of rats.

## 4. Discussion

### 4.1. Weight Gain

In the current study, young Sprague–Dawley male rats were fed diets containing bilberries or cellulose, corresponding to 7% fiber inclusion, in a low- or high-fat feeding regimen for 8 weeks. HFD fed animals did not gain more weight than animals fed LFD, nor did they accumulate additional visceral epididymal white adipose tissue or subcutaneous inguinal white adipose tissue in the body, representing the obese phenotype. Accordingly, studies have demonstrated naturally occurring ‘obesity-resistant’ rats from both Sprague–Dawley and Wistar strains in comparison to their ‘obesity-prone’ counterparts. In those studies, it was shown that rats consuming similar amounts of high-fat diets and exhibiting the same levels of energy absorption, had lower levels of lipid metabolites [26,39] and glucose in blood [8,25,28] and altered liver anabolic features [26]. Other possible reasons for the similar weight gain include the young age of the animals, which might have a varied metabolism and sensitivity to the diets [26,28], and a possible resilience of the gut microbiome [40]. Another interesting finding was that rats fed HFD + BB gained less weight than all the other diet groups. Similar results have been shown in other studies in mice and rats fed high-fat diets with prebiotic blueberry or bilberry that had a lower weight gain compared with diets containing no berries [11,30,41,42]. It may be linked with gel-forming dietary fiber resulting in extra fecal excretion of protein and fat, as described in rats, as well as in human subjects [42,43]. We monitored the health status of all rats throughout the experiment, and there was no sign of diarrhea or behavioral abnormality. Moreover, the diet containing HFD + BB ameliorated the total serum cholesterol levels in the rats.

### 4.2. Expression of Genes

HFD induced a higher liver index and an obvious hepatic steatosis. Interestingly, dietary bilberries almost completely counterbalanced these detrimental effects of HFD. Non-alcoholic fatty liver disease is one common trait during the onset of obesity and might also be a subclinical sign in non-obese subjects [9]. However, expressions of central genes in liver fatty acid uptake and β-oxidation, *Pparγ*, *Pparα* and *Cpt1α*, were unaltered by treatments in the current study. The transcriptional levels of lipogenesis and fatty acid transport, i.e., *Acc*, *Fasn*, *Fabp5* and *Mttp* expression were reduced by HFD independently of bilberry inclusion, suggesting a lower *de novo* lipogenesis activity in the liver by high-fat feeding. In addition, *Scd-1* was regulated in a similar way except for the HFD+BB group, where the liver showed very low expression levels compared to other treatments. This is interesting, since Scd-1 is a central enzyme involved in triglyceride biosynthesis and has been associated with diseases like diabetes, cardiovascular disease and steatosis [44,45]. Another study demonstrated that omega-3 fatty acid deficiency exacerbated antipsychotics induced-hepatic steatosis, showing increased Scd-1 expression, enzymatic activity and triglyceride biosynthesis in the rat liver [46]. Furthermore, different types of prebiotic berries have earlier been demonstrated to reduce triglyceride biosynthesis and Scd-1 expression in high-fat fed mice [41,47].

### 4.3. IBAT and EWAT

Bilberry in the low-fat setting (LFD + BB) reduced eWAT distribution in the rats compared to those fed LFD control. Furthermore, iBAT accumulated more in rats from the HFD + BB group compared to all the other groups, coinciding with the highest expression of *Ucp-1*. This protein is restricted to adipose tissue, where it provides a mechanism for adaptive thermogenesis [48]. The energy expending brown adipocytes dissipate chemically bound energy via Ucp-1 upregulation and produce heat as the end product [49,50,51,52]. This might explain the reduced weight gain and lipid depot in the liver and in the circulation of rats fed HFD + BB in our study, indicating a role of bilberry in regulating adipocytes activity. Another possibility is the increased formation of SCFAs that affects the microbiota composition and also the bile acid composition, as seen in rats fed barley fiber [53]. In addition, it has been shown that berry polyphenols, present in bilberries, induce adipose tissue browning associated with AMP kinase activation, and in parallel modulate gut microbiota in rats and in mice [54,55]. The bioavailability of different polyphenols/anthocyanins is of critical importance. Notably, a previous study reported that soluble bilberry fraction gave rise to the highest amounts of anthocyanins compared to raspberry and blackcurrant, which resulted in highest amounts of SCFAs and lowest liver cholesterol level in rats [32]. Jakobsdottir et al. (2014) suggest that the anthocyanins present in the soluble fractions of bilberries are rapidly metabolized and absorbed in the small intestine and/or fermented by the microbiota in the large intestine. This together with results in the present study, suggest a downstream beneficial effect with prebiotic components.

### 4.4. Gut Microbiota

Clear associations have been made between the host physiology of adipose tissue, hepatic lipid accretion and changes of bacterial community, highlighting the relevance of gut microbiota [3,9,56]. In this study, there were shifts in the four major phyla of rat cecal microbiota including *Firmicutes, Bacteroidetes*, *Actinobacteria* and *Proteobacteria* in response to different diets. The bacterial community diversity and the relative abundance of *Bacteroidetes* decreased, and *Firmicutes* and *Proteobacteria* increased in HFD fed rats, showing a similar representation as previously described in obese and HFD disrupted microbiome [6,7,11,15,27]. Notably, HFD + BB treatment also increased *Proteobacteria* population in the cecal microbiome to an even higher level as with HFD but restored the microbiota diversity and the ratio between *Firmicutes* and *Bacteroidetes* populations. Furthermore, a marked increase in the abundance of *Actinobacteria* was found in the cecum of rats fed HFD (1.2%) and HFD + BB (6.2%) compared with LFD (0.45%) and LFD + BB (0.69%), showing a 10 fold change between treatments. Accordingly, one of the major genera from this bacterial phylotype *Bifidobacterium* sp. was analyzed and exhibited a similar pattern of relative abundance in response to diets. The *Bifidobacterium* sp. abundance has been positively correlated with improved glucose tolerance and mucosal barrier function, and negatively correlated with weight gain and intestinal endotoxin levels in animal models of obesity [16,17,57]. The anti-obesogenic potential of *Bifidobacterium* spp. is also suggested in human patient studies, showing a negative correlation between the incidence of metabolic diseases and bacterial presence [5]. More importantly, prebiotics such as oligofructose that promotes *Bifidobacterium* growth selectively, quenches these high-fat associated changes. In contrast to previously described *Bifidobacterium* reduction by high-fat feeding or obesity [17,57,58], rats fed high-fat diets (HFD and HFD + BB) in our study showed a higher relative abundance of *Bifidobacterium* compared to the low-fat diets, in parallel with an ‘obesity-resistant’ phenotype. An additional factor that regulates host metabolism from gut microbiota may be bacterial cross feeding [18]. This effect has been demonstrated directly by coculture of *Bifidobacteria* and butyrate-producing bacteria. In a human volunteer study of prebiotics, a significant increase of *Bifidobacterium* species and *Faecalibacterium prausnitzii* (a common butyrate-producing bacteria) was reported [13]. In a study of high-fat diet induced diabetes in mice, it was also shown that the *Bifidobacterium* sp. decrease was accompanied by a decrease in the abundance of *Eubacterium rectale* (another common butyrate-producing bacterial species) [17]. In our study, the relative abundance of the butyrate-producing bacteria from rat cecal microbiota was maintained from the HFD group, hinting a role of both these bacteria in withstanding obesity.

### 4.5. Butyrate and Microbiota

Butyrate-producing bacteria represent a functional group that is phylogenetically diverse in the microbial community of the human large intestine. The most common butyrate-producing bacteria are related to *Eubacterium rectale/Roseburia* spp. and to *Faecalibacterium prausnitzii* [14,18,59], but also include *Anaerotruncus* sp. *Coprococcus* sp. and *Anaerostipes* sp., detected in the current study. Given the fact that butyrate-producing bacteria are metabolically versatile [18,59,60], their populations appear to be sensitive to changes of dietary components, particularly dietary fiber substrates. In the present study, bilberry favored the growth of cecal butyrate-producing bacteria, under both LFD and HFD conditions. It is also in agreement with the increased amount of butyrate in the cecum and in the circulation, indicating a stronger bilberry effect overriding the effect of high fat. Furthermore, the different signatures of SCFAs in response to diet changes in the current study revealed that dietary bilberry first exhibit a robust effect by enhancing total SCFAs, acetate and butyrate in both cecum and circulation, under both LFD and HFD conditions. Secondly, HFD induced a total SCFA reduction only locally in the cecum comprised by all three major SCFAs, without alterations in serum. Finally, HFD also reduced the propionate amount in circulation where bilberry could not salvage it. This may partly explain why we found greater response of HFD treatment in the liver rather than in the whole body of rats, as propionate is an important contributor to the liver gluconeogenesis and lipid metabolism [9,18]. One study using direct addition of acetate and propionate to diets with low or high fat inclusion demonstrated that SCFAs improved hepatic lipid metabolism and insulin sensitivity independent of overall obesity phenotype [61]. They also suggested that the presence of propionate plays a key role in the physiological effects of SCFAs on the liver, rather than the ratio of acetate to propionate. Furthermore, the antisteatotic hepatic effects of butyrate have been demonstrated in HFD fed mice treated with tributyrin, a glycerol-ester containing butyric acid [24] and at the same time alleviate HFD induced obesity and inflammation. Nevertheless, our data suggest that prebiotic bilberry may be used to optimize SCFA formation prior to or during the onset of obesity, with an emphasis on butyrate and butyrate-producing bacteria modulation.

## 5. Conclusions

The present study demonstrated that rats fed diets containing bilberry counteracted pre-obesity events including increased liver index, hepatic steatosis, cecal microbiota dysbiosis, enhanced serum total cholesterol levels, whereas iBAT distribution was stimulated and *Ucp-1* was upregulated. Finally, bilberry exerts its prebiotic effects by modifying the SCFA profile in both cecum and circulation, with a special promotion on cecal butyrate formation and butyrate-producing bacteria. This was accompanied by a *Bifidobacterium* sp. increase, which together contributed greatly to the host metabolism homeostasis and achieved a slim final body weight of rats fed on HFD + BB. The fact that HFD alone could not disrupt *Bifidobacterium* sp. and/or butyrate-producing bacteria population may be linked to a ‘resistant’ intestinal barrier function, and an ‘obesity-resistant’ phenotype, which warrant further investigations.

## Figures and Tables

**Figure 1 nutrients-11-01350-f001:**
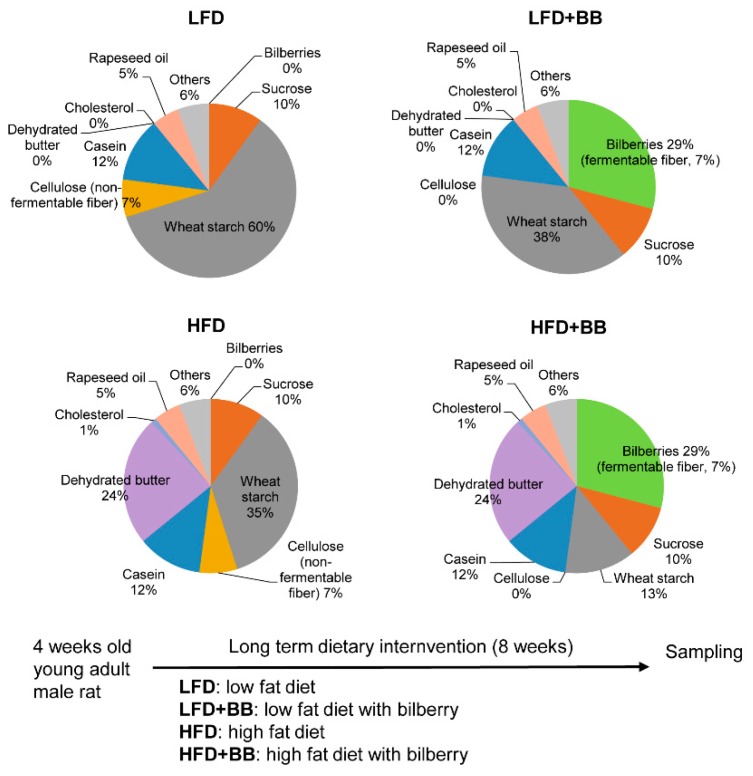
Composition of the four diets employed in the study (i.e., low-fat diet, LFD; LFD with bilberry, LFD + BB; high-fat diet, HFD; and HFD with bilberry, HFD + BB) shown as percentage (detailed additives such as vitamins and minerals are not shown) and scheme of feeding strategy. For diets with bilberry inclusion (LFD + BB and HFD + BB), 29% of bilberry inclusion corresponds to 7% of dietary fiber content, on dry weight basis. All values of nutrients included are on dry weight basis.

**Figure 2 nutrients-11-01350-f002:**
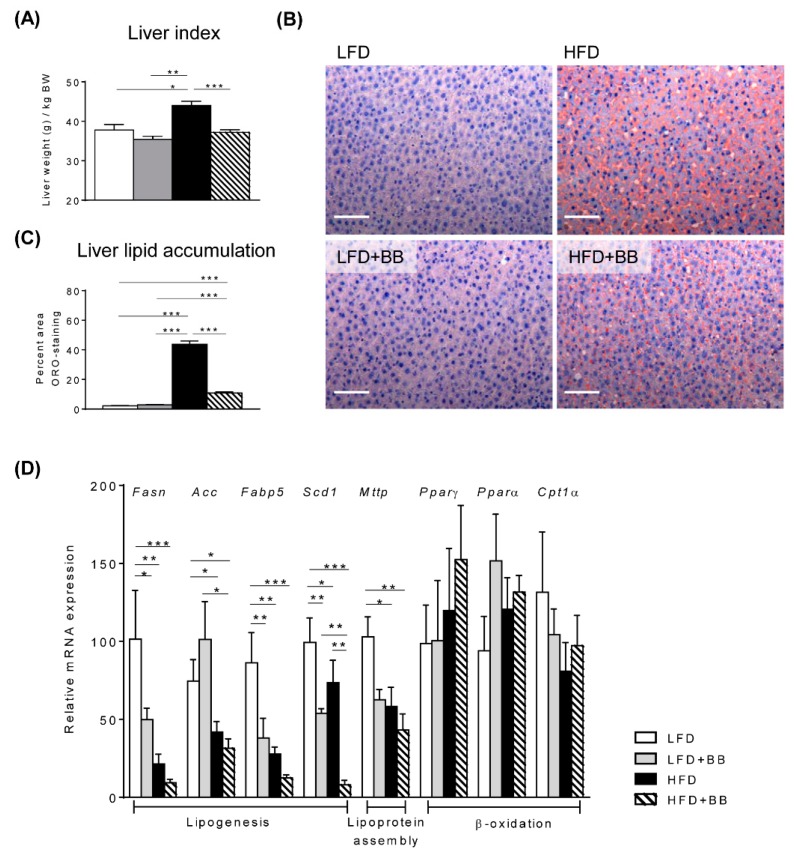
Liver index, lipid accumulation and lipid metabolism gene expression. Rats were fed on low-fat diet (LFD), high-fat diet (HFD), LFD with bilberry (LFD + BB) and HFD + BB, respectively. (**A**) Liver index was calculated as liver weight (g) per kg body weight (BW). (**B**) Representative images of liver sections stained with Oil-red O and hematoxylin. (**C**) Liver lipid accumulation of neutral lipids. The Oil-red O (ORO)-positive area was normalized to per image area. (**D**) Gene expressions related to lipogenesis (*Fasn, Acc*, *Fabp5*, *Scd-1*), lipoprotein assembly (*Mttp*) and lipid β-oxidation (*Pparγ*, *Pparα* and *Cpt1α*) in the liver were analyzed using real-time qPCR. The mRNA expression of samples from the LFD group was set as 100. Values are means ± SEM (stand error of the mean, n = 6–8). Statistical analysis was performed with ANOVA followed by Tukey’s post-hoc test. * *p* < 0.05, ** *p* < 0.01, *** *p* < 0.001.

**Figure 3 nutrients-11-01350-f003:**
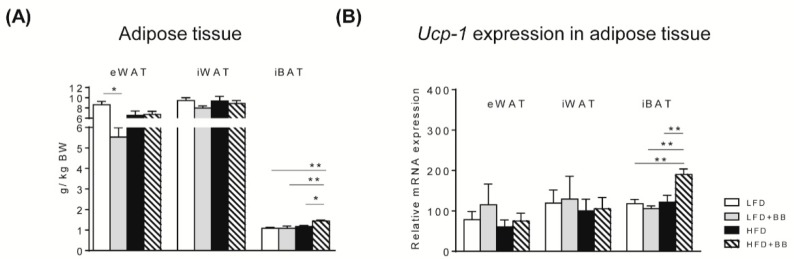
The weight of adipose tissues and uncoupling protein 1 (*Ucp-1*) expression of rats in response to experimental diets. (**A**) The weight (g per kg BW) of adipose tissues from three different compartments of rats: visceral epididymal white adipose tissue (eWAT), subcutaneous inguinal white adipose tissue (iWAT) and interscapular brown adipose tissue (iBAT). (**B**) *Ucp-1* mRNA expression in adipose tissues was investigated by real-time qPCR. Relative mRNA expression was normalized using mean of LFD. Values are means ± SEM (n = 6–8). Statistical analysis was performed with ANOVA followed by Tukey’s post-hoc test. * *p* < 0.05, ** *p* < 0.01.

**Figure 4 nutrients-11-01350-f004:**
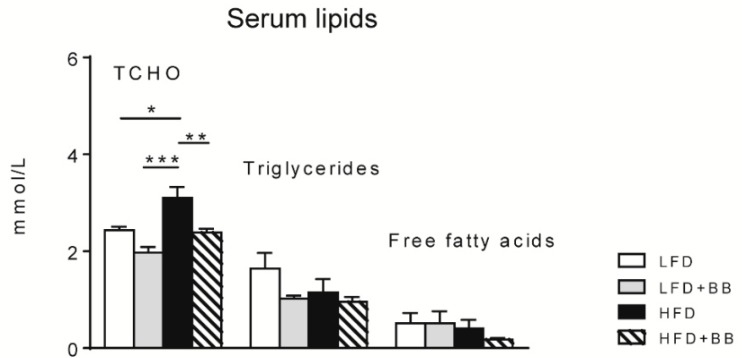
Serum total cholesterol (TCHO), triglycerides and free fatty acid concentration (mmol/L) in rats fed the experimental diets. Values are means ± SEM (n = 7–8). Statistical analysis was performed with ANOVA followed by Tukey’s post-hoc test. * *p* < 0.05, ** *p* < 0.01, *** *p* < 0.001.

**Figure 5 nutrients-11-01350-f005:**
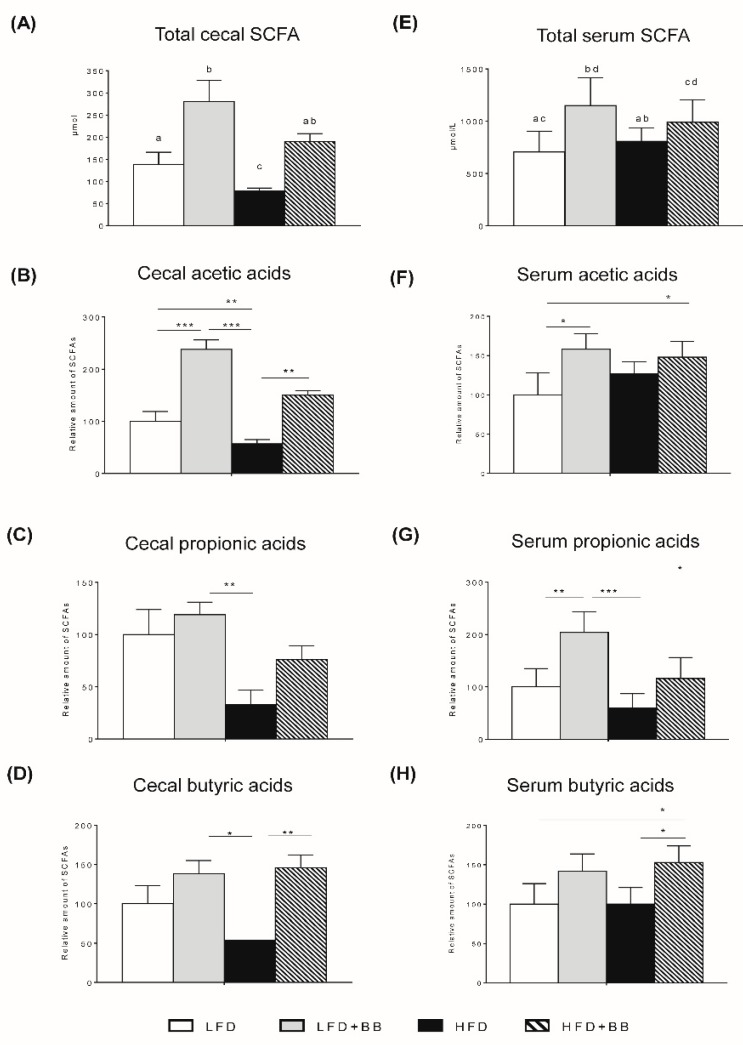
Short-chain fatty acid (SCFA) formation in cecal content and portal vein serum of rats in response to experimental diets. (**A**) Pools of SCFAs in cecal content (µmol) and the relative amount of acetic, propionic and butyric acids in rats fed LFD + BB, HFD and HFD + BB (**B**–**D**). The serum concentration of total SCFAs (µmol/L) and the relative amount of acetic, propionic and butyric acids in rats fed LFD + BB, HFD and HFD + BB (**F**–**H**). Values are means ± SEM (n = 8). ANOVA with Contrast analysis was performed for total SCFAs. Different letters indicate significance (**A**,**E**). Statistical analysis was performed with ANOVA followed by Tukey’s post-hoc test (B-D, F-H). * *p* < 0.05, ** *p* < 0.01, *** *p* < 0.001.

**Figure 6 nutrients-11-01350-f006:**
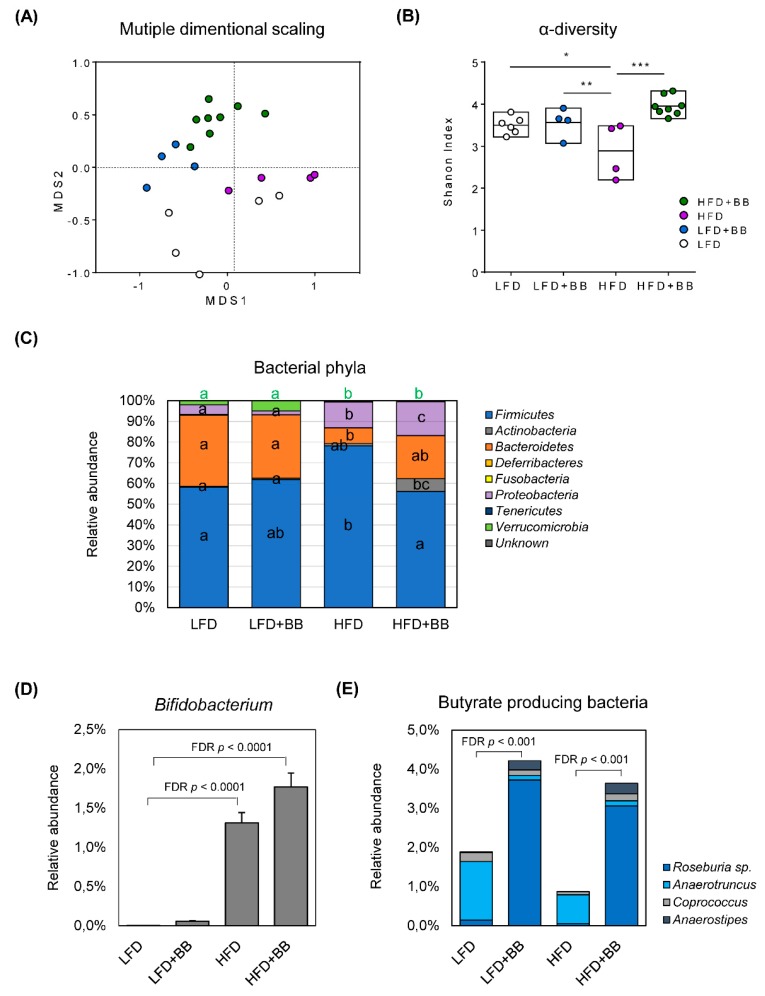
Gut microbiota community of rats in response to experimental diets. (**A**) Microbiota composition in cecal sample was assessed by 16S rDNA analysis with multidimensional scaling analysis, where each symbol represents an individual sample. (**B**) Bacterial community diversity (represented by Shannon index at genus level) (**C**) Bars representing each diet group, show the distribution of bacterial phyla in the microbiota. Values are mean relative abundance of each bacterial phylum (n = 4–8). Significance was determined by using ANOVA followed by Tukey’s post-hoc test and corrected for multiple comparisons with false discovery rate control (FDR). Different letters indicate significance. (**D**,**E**) The relative abundance of *Bifidobacterium* sp. and butyrate-producing bacteria of cecal microbiota (n = 4–8).

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
