# Peer review of "Dietary Fiber in Bilberry Ameliorates Pre-Obesity Events in Rats by Regulating Lipid Depot, Cecal Short-Chain Fatty Acid Formation and Microbiota Composition"

_nutrients, 2019, doi:10.3390/nu11061350_

Reviewer 1 Report

The authors showed that the administration of Bilberry to HFD-fed rat improved NAFLD and altered gene expression in BAT with the changes of intestinal microbiota. They found plausible data to explain the phenotypes induced by Bilberry, but they did not reveal any mechanism between them.

1) Because the authors showed the increased expression of UCP-1 in BAT, they should examine an energy expenditure and respiratory rate by a metabolic cage and a locomotor activity.

2) To dissect a mechanism of increased UCP-1 expression with the alteration of microbiota by Bilberry, please investigate changes in levels and compositions of bile acid in intestine, liver, and serum. The bile acid is known to increase energy expenditure through BAT (Diabetes Obes Metab 19(9): 1214, 2017)

3) Many papers have shown that NAFLD is induced by intestinal microbiota through an increased permeability, so-called “leaky gut” (J Cell Biochem 120(3): 2713-2720, 2019). Please investigate LPS level in sera or a level of fluorodye after the administration to evaluate the permeability of the intestine.

Author Response

Comments and Suggestions for Authors (1)

The authors showed that the administration of Bilberry to HFD-fed rat improved NAFLD and altered gene expression in BAT with the changes of intestinal microbiota. They found plausible data to explain the phenotypes induced by Bilberry, but they did not reveal any mechanism between them.

We have tried to add some possible mechanism in the discussion and kept all changed tracks

1) Because the authors showed the increased expression of UCP-1 in BAT, they should examine an energy expenditure and respiratory rate by a metabolic cage and a locomotor activity.

To examine the energy expenditure and respiratory rate in metabolic cages would have been interesting. However, earlier studies using rats placed in metabolic cages demonstrate that the excretion of fat (and protein) increases considerably (three times) with gel-forming types of fiber such as Vi-Siblin and Inolaxol (Nyman M, Asp N-G, Scand J Gastroenterol 1985, 20, 887-895). This has also been shown in man (Beyer PL, Flynn MA Effects of high- and low fiber diets on human feces, J Am Diet Assoc, 1978, 72, 271-277). Bilberry contains high proportions of soluble and gel-forming types of fiber that also absorbs water (in contrast to cellulose fiber in LFD control that are highly insoluble), which resulted in higher cecum weights (Figure S1) and most probably also fecal weights. Although not that extensively studied, it is quite well known that soluble dietary fiber increases fecal bulk. It is thus likely that the total amount of fecal fat (and perhaps protein) was higher with HFD+BB than with HFD, which may partly explain the difference of energy expenditure between treatments.

2) To dissect a mechanism of increased UCP-1 expression with the alteration of microbiota by Bilberry, please investigate changes in levels and compositions of bile acid in intestine, liver, and serum. The bile acid is known to increase energy expenditure through BAT (Diabetes Obes Metab 19(9): 1214, 2017)

Although an interesting question from the reviewer, the same is valid as for the suggestion above. The experiment is finished, and we cannot start to do further analyses now, since there is no material left for such analyses. We have bile acid analyses for both blood and feces/cecum available, but they require quite much material to be able to do a quantitative determination. Furthermore, this was out of our scope in this investigation. However, it is a good idea to be discussed as a mechanism, since bile acid composition is highly dependent on the microbiota (or vice versa) and the short-chain fatty acids formed by the microbiota from bilberry fiber. We have included some sentences in the discussion (weight gain).

3) Many papers have shown that NAFLD is induced by intestinal microbiota through an increased permeability, so-called “leaky gut” (J Cell Biochem 120(3): 2713-2720, 2019). Please investigate LPS level in sera or a level of fluorodye after the administration to evaluate the permeability of the intestine.

This is indeed very interesting and is exactly what we have assessed in another study focusing on intestinal mucus layers and inflammation, which is now under publication. That study revealed that there was no induction of bacterial counts in the firmly adherent mucus samples in HFD rats, indicating that the barrier was intact to bacteria and bacterial components. However, whether bacterial metabolites penetrated to larger extent could not be excluded, as seen with the changes of serum lipids and serum SCFAs in the current paper. In this respect it is interesting to mention that postprandial glucose levels and appetite responses are lower 15 hours after intake of a fibre rich breakfast compared to a breakfast low in dietary fibre. This is ascribed to changes in microbiota composition and formation of short-chain fatty acids. This has been added in the introduction.

Reviewer 2 Report

The article is interesting, but I have some remark. What was a source of bilberry fiber? How the % of fiber was evaluated? Please add some characteristic of this fiber, what is so special/ different ?

If the whole bilberries or bilberries extract was given to the animal, the content (and activity!) of polyphenols should be taken under consideration.

Author Response

Comments and Suggestions for Authors (2)

The article is interesting, but I have some remark.

What was a source of bilberry fiber? How the % of fiber was evaluated? Please add some characteristic of this fiber, what is so special/ different?

If the whole bilberries or bilberries extract was given to the animal, the content (and activity!) of polyphenols should be taken under consideration.

Fresh frozen bilberries (Vaccinium myrtillus often called wild blueberries) from Kiviks Musteri  (a company in Sweden that are producing fruit juices) were used in the study. The bilberries were freeze-dried before dietary fiber was analysed with an AOAC methodology applying the traditional definition “Dietary fiber is composed of carbohydrate polymers with ten or more monomeric units, which are not hydrolyzed by the endogenous enzymes in the small intestine of humans”. (Prosky LAsp NGSchweizer TFDeVries JWFurda I. J Assoc Off Anal Chem. 1988, 71, 1017-23. Determination of insoluble, soluble, and total dietary fiber in foods and food products: interlaboratory study.) This is a reliable and adequate methodology in many fruits, such as bilberries, since they mostly contain minor amounts of resistant starch and indigestible oligosaccharides. Of practical reasons, i.e. to be able to give the bilberries in a dry formula to the rats, the bilberries had to be freeze-dried, so we did not use an extract of the bilberries. Freeze-drying is further a mild process, keeping the functional properties of the dietary fiber as they are in the original product. After freeze-drying the bilberries were milled and dietary fiber was analysed as described above.

Generally, the dietary fiber composition also is analysed by using gas-chromatography on the fibre residue obtained from the enzymatic-gravimetric methodology. (Theander OAman PWesterlund EAndersson RPettersson D. J AOAC Int. 1995, 78, 4, 1030-44. Total dietary fiber determined as neutral sugar residues, uronic acid residues, and Klason lignin (the Uppsala method): collaborative study.) This has not been done on this specific batch but in another study on the same type of bilberries (Vaccinium myrtillus) having the same content of dietary fiber. In that study we also investigated the content of anthocyanins present in the insoluble and soluble portion of bilberries, since these components are very much discussed in relation to the metabolic syndrome (Jakobsdottir G, Nilsson U, Blanco N, Sterner O, Nyman M. Effects of Soluble and Insoluble Fractions from Bilberries, Black Currants, and Raspberries on Short-Chain Fatty Acid Formation, Anthocyanin Excretion, and Cholesterol in Rats. J Agric Food Chem 2014, 62, 4359-4368). Insoluble and soluble portions of bilberries contributed with 4-17 times more anthocyanins than corresponding fractions of black currants and raspberries. The main dietary fiber components were xyloglucans in both the soluble and insoluble fraction, but those in the soluble fraction were more branched and contained comparatively more galactose and to some extent also arabinose. The soluble and insoluble fiber was highly degraded by the microbiota.

A part of this, fiber procedure has been included in the text in section 2.1.1. Furthermore, a sentence has been added about the content of polyphenols, which mainly are the anthocyanins in bilberries.

Round  2

Reviewer 2 Report

I have only a minor remark: please add as a separete sentence the presence and quantity of antocyanins. Please add the metod of determination.

I the discussion part please discusse the potential role of antocyanins.

Author Response

Reviewer 2_round 2:

I have only a minor remark: please add as a separete sentence the presence and quantity of antocyanins. Please add the metod of determination.

In the discussion part please discusse the potential role of antocyanins.

Reply:

Revised accordingly in method section. Line 130-134.

Revised accordingly in discussion. Line 398-403.